# Genome-Wide Analysis and the Expression Pattern of the MADS-Box Gene Family in *Bletilla striata*

**DOI:** 10.3390/plants10102184

**Published:** 2021-10-14

**Authors:** Ze-Yuan Mi, Qian Zhao, Chan Lu, Qian Zhang, Lin Li, Shuai Liu, Shi-Qiang Wang, Zhe-Zhi Wang, Jun-Feng Niu

**Affiliations:** National Engineering Laboratory for Resource Development of Endangered Crude Drugs in Northwest China, Key Laboratory of the Ministry of Education for Medicinal Resources and Natural Pharmaceutical Chemistry, College of Life Sciences, Shaanxi Normal University, Xi’an 710119, China; mizeyuan@snnu.edu.cn (Z.-Y.M.); triumph@snnu.edu.cn (Q.Z.); luchan@snnu.edu.cn (C.L.); zq182568@snnu.edu.cn (Q.Z.); shidalilin@snnu.edu.cn (L.L.); liushuai.28@163.com (S.L.); wsq@snnu.edu.cn (S.-Q.W.)

**Keywords:** *Bletilla striata*, MADS gene family, expression pattern

## Abstract

*Bletilla striata* (Thunb. ex A. Murray) Rchb. f., a species of the perennial herb Orchidaceae, has potent anti-inflammatory and antiviral biological activities. MADS-box transcription factors play critical roles in the various developmental processes of plants. Although this gene family has been extensively investigated in many species, it has not been analyzed for *B. striata*. In total, 45 MADS-box genes were identified from *B. striata* in this study, which were classified into five subfamilies (Mδ, MIKC, Mα, Mβ, and Mγ). Meanwhile, the highly correlated protein domains, motif compositions, and exon–intron structures of *BsMADSs* were investigated according to local *B. striata* databases. Chromosome distribution and synteny analyses revealed that segmental duplication and homologous exchange were the main *BsMADSs* expansion mechanisms. Further, RT-qPCR analysis revealed that *BsMADSs* had different expression patterns in response to various stress treatments. Our results provide a potential theoretical basis for further investigation of the functions of MADS genes during the growth of *B. striata*.

## 1. Introduction

MADS-box transcription factors are important regulators that participate in several plant growth processes, seed development, and in response to abiotic stress [1,2,3,4,5,6]. MADS-box genes have been identified in plants [7], animals [8], and fungi [9], and are well-known in the ABCDE model, which describes their functionality in the determination of floral organs [10,11]. Additionally, MADSs play essential roles in gymnosperms and mosses without flowers [12,13]. The term MADS is an abbreviation consisting of the initials of *Saccharomyces cerevisiae* transcription factor MCM1 [14], the flower homologous gene *AGAMOUS* (*AG*) of *Arabidopsis thaliana* [15], *DEFICIENS* (*DEF*) of *Antirrhinum majus* [16], and the human serum response factor (SRF) [17].

All MADS-box gene family members contain a DNA-binding domain of ~60 amino acids, known as the MADS-box domain, located at the N-terminal region of the protein [18]. In *A. thaliana*, the gene family can be sub-divided into two classes based on gene structure: type I (Mα, Mβ, Mγ, and Mδ) and type II (MIKC) [19]. This diversity is primarily caused by a gene duplication event [20]. The plant type II genes contain highly conserved MADS domains—of which the K-domain is essential for functional diversity and protein–protein interactions [21]—while type I MADS-box genes have a relatively simple structure and lack a K-domain. Although the MADS-box genes in type II have been well documented and extensively studied, few reports exist that describe type I MADS-box genes in plants [22,23,24].

*Bletilla striata* (Thunb.) Rchb. f. is a perennial medicinal herb of the Orchidaceae family that is extensively distributed across China, Korea, Japan, and Myanmar [25]. According to the earliest pharmacopeia of traditional Chinese medicine, *Shennong’s Materia Medica Classic*, *B. striata* has a long history of treating tuberculosis, whooping cough, hemorrhoids, eye diseases, and more [26]. *B. striata* also exhibits a range of unique flower shapes with high ornamental value. MADS-box transcription factors are key regulators of flower development, which might improve the quality of *B. striata* by expediting flowering times.

The development of bioinformatics technology has resulted in more specific plant genomics research, especially concerning the functional screening and verification of genes. Examples of these research findings include: *PIN-FORMED* (*PIN*) genes and *brassinazole-resistant* (*BZR*) genes have a vital role in *Triticum aestivum* L. growth and development under various stress conditions [27,28]; the KCS gene family in barley shows diverse expression patterns under drought stress [29]; LncRNAs in *Capsicum annuum* regulate transcription factors by interacting with miRNAs [30]; natural antisense transcripts in *Salvia miltiorrhiza* potentially regulate the biosynthesis of bioactive compounds [31]; and that *AT-hook motif nuclear-localized* genes mediate the stress response of soybean [32]. The MADS-box gene family has also been studied in many plants, especially *A. thaliana* [7,19,33] and rice [1,34,35], as well as orchids such as *Erycina pusilla* [36] and *Phalaenopsis* orchid [37]. The identification of MADS-box family genes in orchids can provide useful information for both breeding and flowering. However, there is no in-depth research on MADS-box genes and their roles in *B. striata*. 

In this study, the first genome-wide analysis of *B. striata* was conducted, including its genetic structure, motif compositions, *cis*-elements, chromosome distribution, and synteny analysis. Based on sequence alignments and phylogenetic tree analyses, we divided the *BsMADSs* into five subfamilies. The expression patterns of the *BsMADSs* revealed that the stress-responsive *BsMADS* genes responded to various treatments. Our results may provide a better understanding of the origins of gene diversification within each subfamily, and the basis for gene function analysis to elucidate their specific roles in the development of *B. striata*.

## 2. Results

### 2.1. Identification of BsMADS

A total of 45 presumed MADS proteins were searched from the *B. striata* protein databases using the BLASTP program with 146 AtMADS (70 M-type MADS and 76 MIKC MADS) proteins as queries. All *B. striata* MADS proteins (designated BsMADS01 to BsMADS45) were confirmed to contain a single SRF-TF domain by InterProscan and SMART (Appendix A). The number of *BsMADS* genes was less than the identified MADS genes in *Arabidopsis* (146) and rice (75). 

The basic features of the BsMADSs were predicted, and the coding sequence (CDS) lengths of 45 *BsMADS* genes ranged from 438 bp (*BsMADS*41) to 8612 bp (*BsMADS*14); thus, the proteins encoded would potentially include from 75 (BsMADS45) to 433 amino acids (BsMADS20). The molecular weight (MW) ranged from 8713.2 to 49280.86 Da. The isoelectric point (pI) ranged from 4.66 (BsMADS23) to 10.43 (BsMADS45). The prediction of subcellular localization revealed that all BsMADS proteins resided in the nuclear region (Appendix A).

### 2.2. Analysis of Multiple Sequence Alignments and Cis-Acting Elements

Multiple sequence alignments were performed based on the conserved SRF-TF domain of BsMADSs (Figure 1). The results revealed some highly conserved amino acids in the SRF-TF domain, including the 15th arginine (R), 21st lysine (K), 25th glutamic acid (E), 26th leucine (L), etc. (Figure 1a), which verified that the domain was a highly conserved DNA-binding/dimerization region [38]. 

To comprehensively investigate the function and regulatory roles of the *BsMADSs*, we analyzed the various *cis*-acting elements of the promoter regions. The identified *cis*-acting elements were classified into five main functional classes: transcription, cell cycle, hormone, abiotic or biotic stress, and development. A total of 103 types of elements were found, most of which were associated with different stress conditions, including hormonal responses such as ABA, GA, MeJA, SA, and auxin, as well as abiotic responses such as wound, low temperature, drought, etc. (Figure 2, Appendix A). We focused on drought inducibility and MeJA-related elements to predict the functions of the *BsMADS* genes.

### 2.3. Phylogenetic Analysis of the MADS-Box Family between B. striata and Arabidopsis

To clarify the phylogenetic relationships of the MADS family proteins between *B. striata* and *Arabidopsis*, all whole-length amino acid sequences of 45 BsMADS and 146 AtMADS were employed to perform a multiple sequence alignment (MSA) protein analysis. An unrooted phylogenetic tree was developed using the neighbor-joining (NJ) method according to the similarity and topology of sequences, and the 45 BsMADS were distributed into five subgroups: A to E (Figure 3). The results could be highly useful for predicting the various roles of unknown BsMADS according to the functionality confirmed in AtMADS or subgroups in *Arabidopsis*, which might contribute to the selection of target BsMADS for further functional analysis.

### 2.4. Gene Structure, Motif Composition, and Ka/Ks Analysis of BsMADSs

To glean insights into the similarities and diversity of gene structures and conserved motifs in BsMADSs, a phylogenetic tree was constructed based on multiple sequence alignments (Figure 4a), which clustered almost identically to the result shown in Figure 3, indicating good consistency. The genomic DNA and CDS sequences were employed to analyze the intron and exon structures of the *BsMADSs* (Figure 4c). The results of the gene structure analysis showed that the number of introns varied from zero to ten. In the same cluster of the phylogenetic tree, most genes possessed similar exon–intron structures, where among them, 26.7% of the *BsMADSs* contained only one exon without introns.

A total of eight conserved motifs were detected in the BsMADSs (Figure 4b), with each motif sequence logo displayed (Figure 4d). Most of the BsMADSs contained Motif-1, Motif-2, and Motif-4. Motif-7 was involved in 19 genes, while Motif-3 existed in 25 genes. Motif-8 was found in five genes, whereas only three genes contained Motif-6. We also found that the MADS genes with close evolutionary relationships shared the same motif compositions, suggesting that the function of MADS proteins is similar.

We discovered that there were 14 pairs of homologous genes in the *BsMADSs* (Figure 4a), which suggested that ~62.2% *BsMADS**s* were replicated and that the BsMADS family had experienced evolutionary gene expansion. To explore the influence of evolutionary factors on the BsMADS family, the Ka/Ks ratios of 14 *BsMADS* gene pairs were computed based on the phylogenetic tree (Figure 4). The results showed that the ratios of 92.9% orthologous gene pairs were no more than 1, which implied that a strong purifying selective pressure existed in the homologous genes during the course of their evolution (Appendix A).

### 2.5. Synteny Analysis and Chromosomal Distribution of MADS Genes

Gene duplication events may have given rise to new functionalities that assisted *B. striata* in adapting to changing environments, thus enriching gene family members. Three comparative synteny maps were constructed, including *B. striata*, *A. thaliana*, *Vanilla fragrans*, and *Vitis vinifera* (Figure 5). A total of 17 MADS genes showed a syntenic relationship with those of *V. fragrans*, followed by *A. thaliana* (4) and *V. vinifera* (10).

To gain insights into the evolution of the 45 *B. striata* MADS-box genes, we analyzed their genomic distribution and discovered that they were unevenly distributed on chromosomes 01–10 and 12–14 (except for five members that were found within repeat sequences or unassembled scaffolds) (Figure 6). Chr01 and Chr03 contained six MADS-box genes, while Chr08, Chr09, and Chr14 had only one. The results of genomic distribution proved that some *B. striata* MADS-box genes of type I or type II were located in the same chromosomal region. A similar previously reported situation regarding the MADS-box genes of *Arabidopsis* [39] and rice [40] suggested that they were widely distributed across the genomes of the common monocotyledon and dicotyledon ancestors.

### 2.6. Expression Profile of BsMADS Genes under Non-Stressed Growth Conditions

To explore the tissue-specific expression of *BsMADS**s*, we examined their expression profile in roots, stems, leaves, and flowers under non-stressed growth conditions based on quantitative real-time PCR (RT-qPCR) analysis (Figure 7, Appendix A). The expression levels of diverse *BsMADS**s* changed significantly between different tissues. From the heatmap, *BsMADS01*, *BsMADS**05*, *BsMADS06*, *BsMADS07*, *BsMADS11*, *BsMADS27*, and *BsMADS40* exhibited a higher transcript accumulation in flowers, and they all belonged to group E (MIKC) (Figure 3). This result verified that the MADS-box genes described in the ABCDE model belonged to the MIKC subclass [41,42].

### 2.7. Analysis of Expression Patterns under Different Stress Treatments

Eight *BsMADS* genes were randomly selected to further analyze their expression level patterns under different abiotic stressors (Figure 8) and hormone treatments (Figure 9) at five time points (0, 1, 3, 6, and 12 h). *BsMADS03* belonged to group A, *BsMADS43* belonged to group B, *BsMADS18* belonged to group C, *BsMADS10* and *BsMADS33* belonged to group D, and *BsMADS01*, *BsMADS07*, and *BsMADS11* belonged to group E. Genetic patterns with significantly different expression levels were detected by RT-qPCR. As shown in Figure 7 and Figure 8, *BsMADS33* had robust responses to all treatments, while the expressions of *BsMADS0**1* and *BsMADS07* were not obvious under the abiotic treatments, and *BsMADS43* was not induced by NaCl. 

The expression levels of two MADS genes (*BsMADS03* and *BsMADS33*) dramatically increased under the CuSO_4_ treatments. Five genes upregulated under AgNO_3_ stress; among them, *BsMADS10* was mostly upregulated ~60-fold at 1 h, which then decreased, and *BsMADS33* continued to increase after being upregulated ~60-fold at 3 h. *BsMADS10* downregulated following the ABA and SA treatments. All genes in the ABA treatment group were initially significantly downregulated and then upregulated, which indicated that the ABA treatment inhibited gene expression within 1 h.

Taking *BsMADS33* stimulated by external ABA as an example, we hypothesized that the phosphorylation of SnRK2 protein kinase caused by the ABA signal promotes the binding of ABRE (ABA response elements) to ABFs (downstream targets) [43], thus regulating the increased gene expression and contributing to the protein products conducive to plant resistance to stress. 

## 3. Discussion

The MADS family is one of the largest plant transcription factor families, playing a critical role in plant growth and development and in the regulation of plant responses to both abiotic and biotic stresses [44,45]. Although this family has been widely researched for many plants, including *Arabidopsis* [4,7], rice [34], maize [46,47], *Erycina pusilla* [36], and *Phalaenopsis* [37], the identification, expression patterns, and functions of MADSs based on the whole-genome sequence of *B. striata* have remained poorly understood. Therefore, we conducted a comprehensive and systematic analysis of the MADS-box gene family in *B. striata*. 

For our study, a total of 45 MADS-box genes were identified in *B. striata*, which was lower than that of both *Arabidopsis* (146 genes) [19] and rice (75 genes) [1], implying that the MADS family had diminished over the course of evolution. The number of genes varied for different species, which may have been caused by gene duplication (e.g., tandem and segmental duplication) [48,49]; however, members of the *RsMADS* may not have undergone such a process [50]. It was also revealed that gene function was a major determinant of gene family size, thus further suggesting that natural stressors were the major driving evolutionary forces [51]. 

To further explore the restrictive conditions during the evolution of the BsMADS family, the Ka/Ks ratios of the 28 paralogous *BsMADS* gene pairs were computed. The smaller the value of Ka/Ks between gene pairs, the more severe the selective constraint they evolved. The Ka/Ks ratios of most duplicated *BsMADS* gene pairs were less than 1, which indicated that the purification selection pressure played a dominant role in the evolutionary process of this family (Appendix A). The results revealed that the MADS-box gene family was currently being purified under *B. striata* selection.

According to the phylogenetic tree between *B. striata* and *Arabidopsis*, the *BsMADS* genes were divided into five subgroups, which also provided a reliable reference for the prediction of functional target *BsMADS* genes (Figure 3). Some BsMADSs were clustered in the same branch as the known AtMADSs, suggesting the conservation of MADS protein functions between the two species. For example, *AGL6* (*AGAMOUS-LIKE6*, AT2G456501) is involved in the development of bractless flowers in *Arabidopsis* [7], while the expression level of *AGL12* contributes to the regulation and development of root structures (*AGAMOUS-LIKE12*, AT1G716921) [52], both of which belong to group E (MIKC). 

Consequently, *BsMADSs* in the MIKC subgroups might be regarded as prime candidate genes for the development of plant organs. The results of the sequence alignments revealed characteristic amino acids (e.g., the majority of the BsMADSs contained one critical amino acid residue, which was the 25th glutamic acid (E)). The secondary structural elements of the SRF-TF domain were found to contain two α-helix and three β-sheets (Appendix A). SRF had been reported to activate transcription by interacting with the B-box [53]. 

Structural analysis determined that 26.7% of the *BsMADSs* contained just one exon (Figure 4). The close *BsMADSs* had similar structures and conserved motif complements, which also confirmed the impregnable evolutionary conservation. Conserved motifs might serve as potential DNA binding sites to regulate the expression of specific genes [54,55]. The transcriptional activities and specificity of DNA combinations depended on unique structures and motifs. Therefore, based on the phylogenetic tree (Figure 3), motif composition analysis (Figure 4b), and gene structures (Figure 4c), we speculated that the *BsMADSs* with the same subgroups might have similar functions, which were highly conserved.

The results of the *cis*-elements analysis suggested the responses of the *BsMADS* genes to various biotic and abiotic stressors. There have been many studies of plant MADS transcription factors involved in plant growth and development. For example, phytohormone abscisic acid (ABA) was observed to inhibit the expression of *FAMADS1a* in strawberries and promote fruit ripening [56]. *TaMADS2* isolated from wheat was useful for the treatment of wheat-stripe rust interactions [57]. The yields of transgenic rice that overexpressed *JCMADS40* were reduced [58]. *CpMADS3* in cyclamen had a novel function in flowering [59]. Furthermore, MADS genes participate in multiple abiotic stress responses [60,61,62]. 

This suggested that *BsMADS* genes were likely to be involved in these physiological processes. To further investigate the functionality of MADS-box genes, we analyzed the expression levels of eight randomly selected *BsMADSs* under different treatments (Figure 8 and Figure 9). *BsMADS07*, *BsMADS10*, and *BsMADS11* had no significant differences following the MeJA treatment, while the expression of the *BsMADS33* gene significantly increased, suggesting that it may regulate the transcription of related genes. Under Cu^2+^ and Ag^+^ stress conditions, both *BsMADS33* and *BsMADS**10* were upregulated and belonged to group D (Mδ). 

Therefore, Mδ MADS genes in *B. striata* may play critical roles in responding to abiotic stresses. According to the phylogeny analysis, the selected *BsMADS* genes exhibited similar stress-responsive expression patterns to model species within the same clade or subgroup. Among them, *BsMADS33* was upregulated under all treatments; thus, we predicted that it could be employed as a candidate gene for later functional research. In brief, the results of our study might provide a credible reference for further research on the functions of type I MADS genes in *B. striata*.

## 4. Materials and Methods

### 4.1. Plant Material and Treatment

In order to perform a genome-wide analysis of *B. striata*, we planted germplasm resources at the greenhouse in the National Engineering Laboratory for Resource Development of Endangered Chinese Crude Drugs, Northwest China. Seeds (2n = 2x = 32) were collected and sprouted on a seedling bed at 25 °C under natural lighting (16 h light/8 h dark) at 60–80% humidity. Two-month-old seedlings were employed for stress-related expression profile analysis. For the abiotic stress treatments, the seedlings were treated with NaCl (200 mM), AgNO_3_ (200 mM), and CuSO_4_ (200 mM) solutions, respectively. The above aseptic seedlings were subjected differently in an MS liquid medium, which contained methyl jasmonate (MeJA, 200 μM), abscisic acid (100 μM), and salicylic acid (10 mM) for the hormonal treatment. The stress samples were collected at 0, 1, 3, 6, and 12 h with three biological replicates. Each sample was instantly frozen in liquid nitrogen and stored at −80 °C for further RNA isolation.

### 4.2. Identification of MADS-Box Gene in B. striata

A total of 146 previously identified *Arabidopsis* MADS-domain protein sequences were used to query the *B. striata* database. Further, a BLASTP search of the *B. striata* database was performed using the conserved MADS-domain HMM profile (PF00319) obtained from the Pfam website (Pfam 32.0, http://pfam.xfam.org/, accessed on 9 October 2021) with an e-value (expected value) cut-off set to 0.01. All potential proteins were confirmed through the SMART service (http://smart.embl-heidelberg.de/, accessed on 9 October 2021). The relative molecular mass and theoretical isoelectric point of all confirmed BsMADS proteins were predicted by ExPASy ProtParam tool (http://expasy.org/, accessed on 9 October 2021).

### 4.3. Multiple Alignment and Phylogenetic Analysis

For multiple alignment analysis, the InterPro program (http://www.ebi.ac.uk/interpro/, accessed on 9 October 2021) was used to acquire the core sequence of the SRF-TF domain, which was further analyzed via the DNAMAN and SMART software programs. The Weblogo (http://weblog.berkeley.edu/logo.cgi, accessed on 9 October 2021) online program was employed to reveal the characteristics of the domain.

MADS protein sequences from *Arabidopsis* were downloaded from the PlantTFDB (http://planttfdb.gao-lab.org/, accessed on 9 October 2021). The ClustalX (1.83) program was utilized to perform the alignment of multiple amino acid sequences. The neighbor-joining (NJ) method was employed to construct a phylogenetic tree using MEGA v 7.0 with 1000 bootstraps and default parameters of ClusterW.

### 4.4. Prediction of Conserved Motifs and Gene Structure Analysis

The online MEME (http://meme-suite.org/tools/meme, accessed on 9 October 2021) program was utilized to search and detect the protein motifs of 45 BsMADSs with expected e-values of less than 2 × 10^−30^ [63]. The following default parameter setting changed the number of motifs into eight. Next, the result of the XML file obtained from MEME was displayed using TBtools v 0.58 [64]. The exon–intron structures of the *BsMADSs* were displayed using GSDS (http://gsds.cbi.pku.edu.cn, accessed on 9 October 2021) [65].

### 4.5. Cis-Elements and Ka/Ks Analysis

The 2500 bp promoter sequences located upstream of the gene start codon were obtained from the whole-genome sequences of *B. striata* using BioEdit software. The potential *cis*-elements of *BsMADSs* were searched in the PlantCARE database [66]. With the purpose of examining whether positive selection existed in the evolution of *BsMADS* genes, the synonymous substitution rate (Ks) and non-synonymous substitution rate (Ka) values of homologous gene pairs, with their amino acid sequences, were calculated by the online software Clustal Omega (https://www.ebi.ac.uk/Tools/msa/clustalo/, accessed on 9 October 2021) and PAL2NAL (http://www.bork.embl.de/pal2nal/, accessed on 9 October 2021) [67].

### 4.6. RT-qPCR Analysis

Four different tissue (root, stem, leaf, and flower) samples of two-year-old *B. striata* were individually collected during its flowering development stage. These samples were used to conduct RT-qPCR to detect the tissue-specific *BsMADSs* expression patterns. The PCR conditions were set at 95 °C for 30 s, 95 °C for 5 s, and 60 °C for 30 s with 45 cycles. Each reaction had three biological and technical replicates, which used 30-fold diluted cDNA as a template. The 2^−^^△△CT^ method was employed to calculate the corresponding expression values of the *BsMADSs*. According to our laboratory studies, we selected *BsGAPDH* as the internal reference gene in this work, with the primer sequences listed in Appendix A.

## 5. Conclusions

In summary, we identified 45 *BsMADS* genes in *B. striata*, and the phylogenetic relationships, gene structures, conserved motifs, *cis*-acting elements, and expression profiles of this family were comprehensively analyzed for the first time. This revealed that *BsMADS* had a significant role in response to ABA, SA, MeJA, NaCl, CuSO_4_, and AgNO_3_ stressors. Based on the analysis of the phylogenetic tree and expression patterns, we predicted and verified the potential functions of the *BsMADSs* in our study, which provided a credible reference for further exploring the regulatory mechanisms and stress resistance in the development of *B. striata*.

## Figures and Tables

**Figure 1 plants-10-02184-f001:**
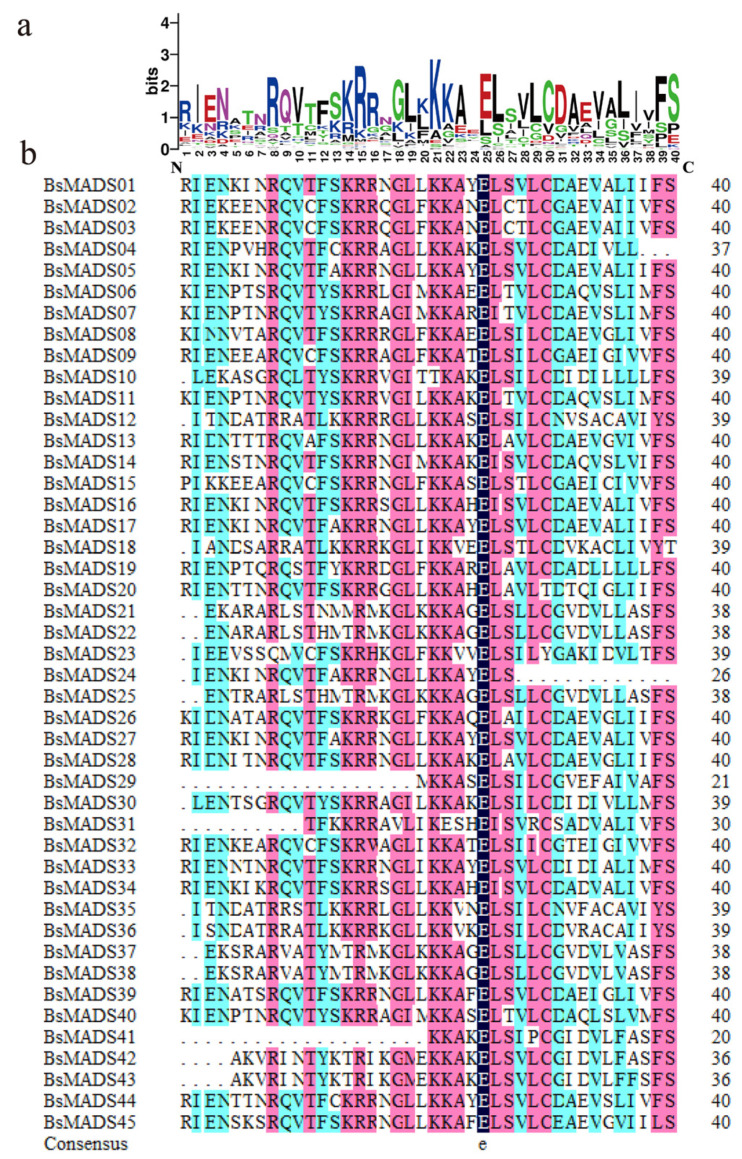
Sequence logo and multiple alignment analysis of the SRF-TF domain: (**a**) sequence logo of the SRF-TF domain. (**b**) Multiple alignment analysis of the SRF-TF domain. The different colors of shading indicate the same and conserved amino acid residues, respectively.

**Figure 2 plants-10-02184-f002:**
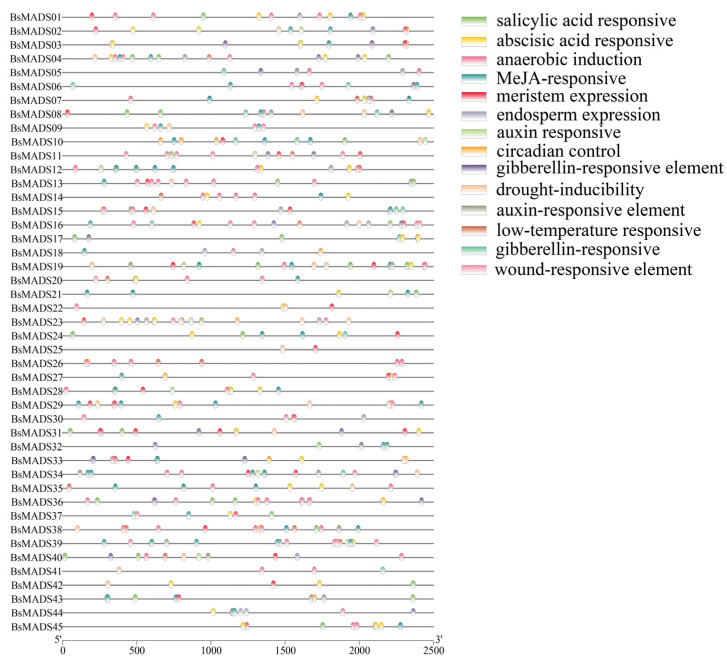
Distribution of various *cis*-acting elements in the promoter regions of *B. striata* MADS genes (*BsMADSs*). The *cis*-elements of the *BsMADSs* were searched using the online PlantCARE website. Special *cis*-acting elements were selected (e.g., auxin-responsive elements), and TBtools was used to create the diagrams.

**Figure 3 plants-10-02184-f003:**
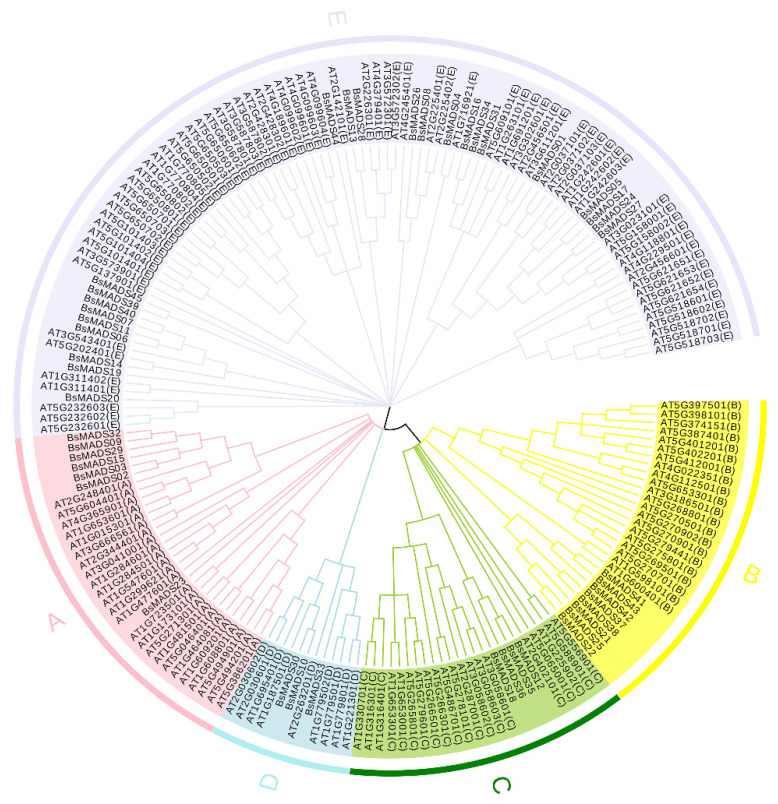
Phylogenetic tree of the MADS-box family proteins in *B. striata* and *Arabidopsis*. The unrooted neighbor-joining (NJ) tree was based on the amino acid sequences of AtMADS from *B. striata* (45) and *Arabidopsis* (146) using MEGA7 with 1000 bootstrap replicates. The names of the (A-E) groups shown outside the circle indicate different MADS subgroups—A (Mα), B (Mβ), C (Mγ), D (Mδ), and E (MIKC); A, B, C, and D groups belonged to type I, and E groups belonged to type II.

**Figure 4 plants-10-02184-f004:**
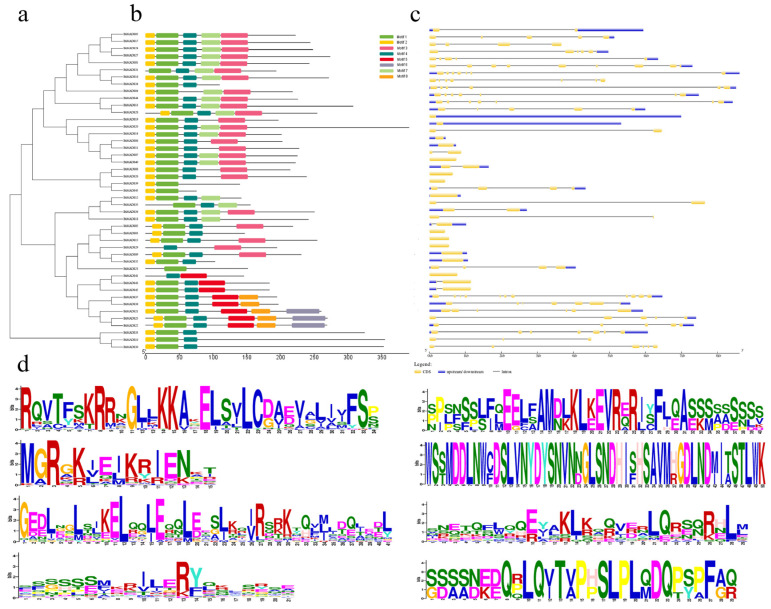
Phylogenetic relationships, conserved motifs, and exon–intron organization of MADS genes: (**a**) the phylogenetic tree contained 45 MADS proteins (named BsMADS01 to BsMADS45) from *B. striata*. (**b**) The motif patterns of the 45 BsMADS proteins. Each motif is shown as boxes of different colors. (**c**) Exon–intron structures of MADS genes from *B. striata*. Exon(s) and intron(s) are represented by yellow boxes and black lines, respectively, while the untranslated regions are represented by blue boxes. The number manifested the phases according to the *BsMADSs* introns. (**d**) Sequence logos of each motif.

**Figure 5 plants-10-02184-f005:**
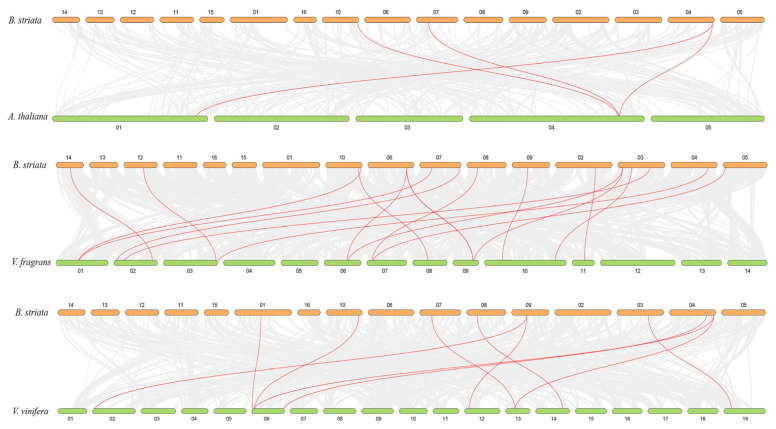
Synteny analysis of *MADS* genes between *B. striata* and three other plants (*A. thaliana*, *V. fragrans*, and *V. vinifera*): gray lines indicate the collinear blocks within the *B. striata* genome and the other genomes, while the red lines indicate MADS gene pairs.

**Figure 6 plants-10-02184-f006:**
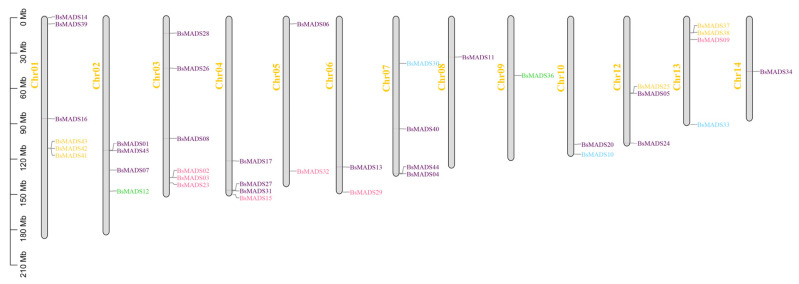
Chromosomal localization of 45 MADS genes in *B. striata* at the megabase (Mb) scale. The scale of five genes (*BsMADS18*, *BsMADS19*, *BsMADS21*, *BsMADS22*, and *BsMADS35*) could not be anchored on a specific chromosome. The pink genes belong to group A (Mα), the yellow genes belong to group B (Mβ), the green genes belong to group C (Mγ), the blue genes belong to group D (Mδ), and the purple genes belong to group E (MIKC).

**Figure 7 plants-10-02184-f007:**
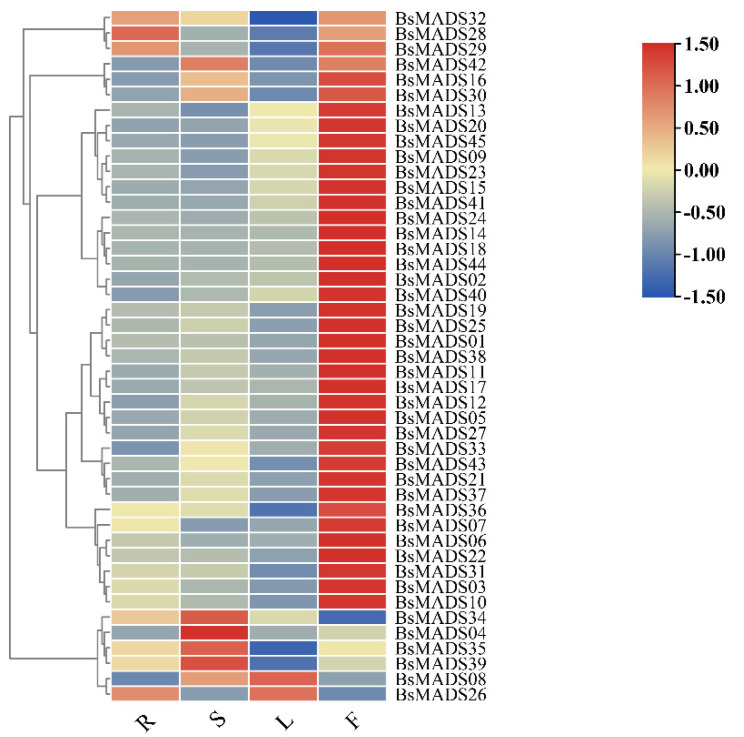
Heatmap showing the expression profile of *BsMADSs* in four different tissues (root, stem, leaf, and flower)—R (root), S (stem), L (leaf), and F (flower), respectively.

**Figure 8 plants-10-02184-f008:**
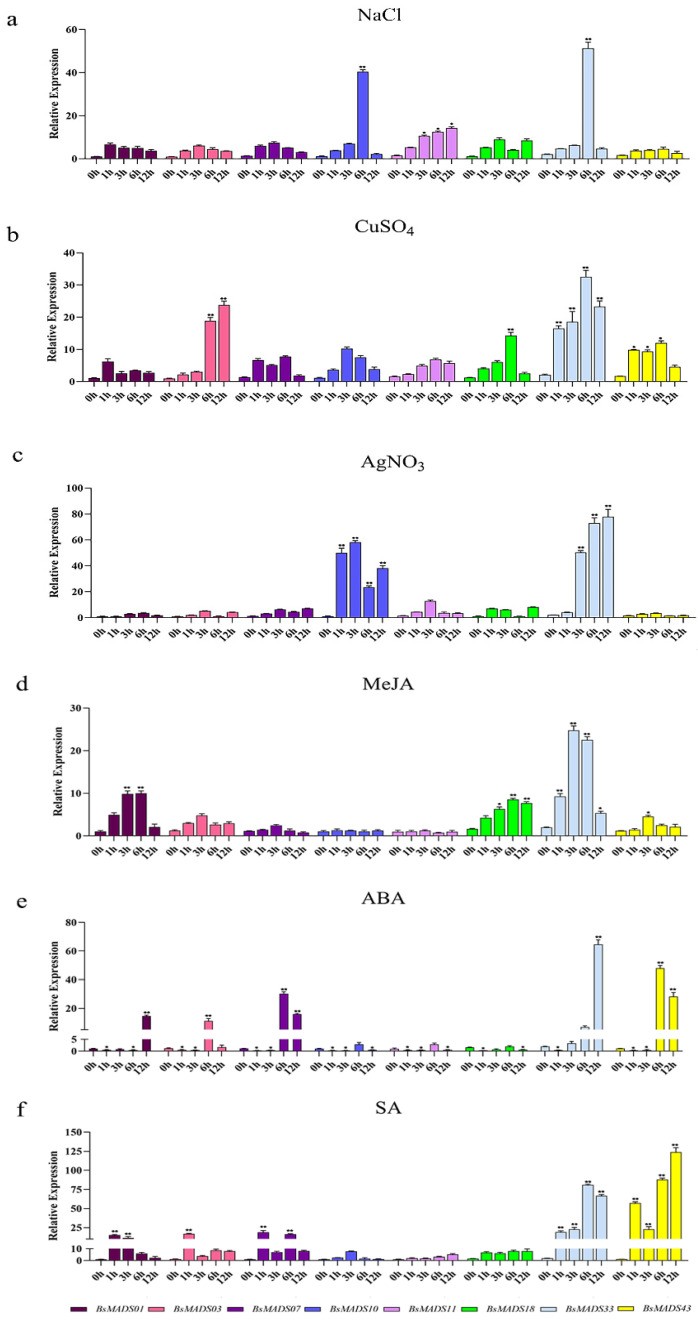
The expression levels of 8 *BsMADSs* in response to various stress treatments at 0, 1, 3, 6, and 12 h: (**a**) NaCl, (**b**) CuSO_4_, (**c**) AgNO_3_, (**d**) MeJA, (**e**) ABA, and (**f**) SA. The data were analyzed via the two-way method using GraphPad Prism, version 8. Asterisks (*p* < 0.05) indicate significant differences compared with the control group (* *p* < 0.05, ** *p* < 0.01). Error bars represent standard error for data obtained in three biological replicates.

**Figure 9 plants-10-02184-f009:**
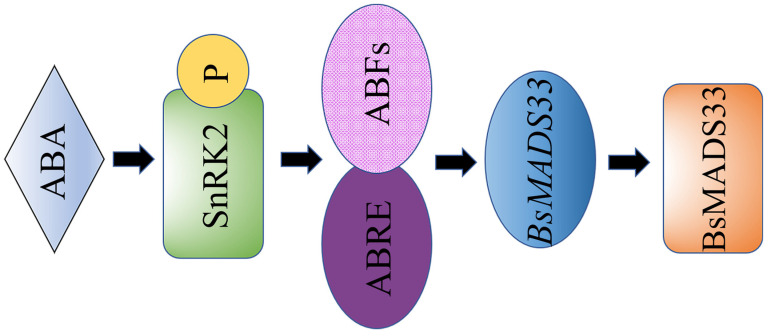
Annotation map of *BsMADS33* hypothetical pathway in response to ABA stress.

## Data Availability

The data used to support the findings of this study are available from the corresponding author upon request.

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
