# Peer review of "Genome-Wide Analysis and the Expression Pattern of the MADS-Box Gene Family in *Bletilla striata"

_plants, 2021, doi:10.3390/plants10102184_

Round 1

Reviewer 1 Report

In this manuscript, the author did the genome-wide analysis and the expression pattern of the mads-box gene family in Bletilla striata. A total of 45 MADS-box genes were identified from B. striata in this study, which were classified into five subfamilies (Mδ, MIKC, Mα, Mβ, and Mγ). Meanwhile, the highly correlated protein domains, motif compositions, and exon-intron structures of BsMADSs were investigated according to local B. striata databases. Chromosome distribution and synteny analyses revealed that segmental duplication and homologous exchange were the main BsMADSs expansion mechanisms. Further, RT-qPCR analysis revealed that BsMADSs had different expression patterns in response to various stress treatments. These results might provide a theoretical basis for the further investigation of the functions of MADS genes during the growth of B. striata.

The manuscript is very well written, but for the betterment of the manuscript, I have few comments to make.

  1. It would be nice if the author tries to functionally validate at least one gene found in this study for any abiotic stress tolerance in B. striata.
  2. Make one figure that depicts a hypothetical mechanism with the genes identified in this study for abiotic stress tolerance.
  3. All the figure quality is poor and very small. Therefore, it is tough to see the letter in the figure. Please enlarge all of them with high resolution.
  4. Enlarge figure 8 with portrait mode.
  5. The introduction is very short and did not include many recent reported genome-wide studies. Please cite the following study: a. Genome-Wide Identification and Characterization of PIN-FORMED(PIN) Gene Family Reveals Role in Developmental and Various Stress Conditions in Triticum aestivum. b. Genome-wide identification and expression pattern analysis of the KCS gene family in barley. c. Genome-wide identification and characterization of abiotic stress-responsive lncRNAs in Capsicum annuum. d. Genome-Wide Identification and Characterization of the Brassinazole-resistant (BZR) Gene Family and Its Expression in the Various Developmental Stage and Stress Conditions in Wheat (Triticum aestivum). e. Genome-wide identification and functional characterization of natural antisense transcripts in Salvia miltiorrhiza. f. Genome-wide identification and expression analysis of the AT-hook Motif Nuclear Localized gene family in soybean.

Change at                                         

L72 deisgnated BsMADS01 to designated BsMADS01.

Manuscript is plagiarized at L32-35, L98-99, L110, L123-124, L128-129, L136-137, L147, L170-171, L290-291, L306-309, L333-334, L341-342, L351-352. Please clean it.

Author Response

Dear Reviewers:

On behalf of my co-authors, we thank you very much for giving us an opportunity to revise our manuscript, we appreciate editor and reviewers very much for their positive and constructive comments and suggestions on our manuscript entitled “Genome-Wide Analysis and the Expression Pattern of the MADS-box Gene Family in Bletilla striata” (ID: plants-1386932). Those comments are all valuable and very helpful for revising and improving our paper, as well as the important guiding significance to our research. We have studied comments carefully and made correction which we hope meet with approval. Revised portion are marked in yellow highlights in the paper. All changes could be found in the “manuscript-marked” file. The yellow highlights show the changes we made according to the reviewer’s comments. We would like to express our great appreciation to you and reviewers for comments on our paper and hope that the correction will meet with approval.

Reviewer 1

  1. It would be nice if the author tries to functionally validate at least one gene found in this study for any abiotic stress tolerance in B. striata.

Response: Thanks for the helpful suggestions. It is really true as Reviewer suggested that we need to do deeper research. Due to the impact of the global COVID-19 pandemic, many experiments cannot be carried out as usual. We are very sorry for the absence of gene function verification in the article, which will become our future research direction. We will pay attention to this gene family in B. striata and share the results if there is any further progress.

  1. Make one figure that depicts a hypothetical mechanism with the genes identified in this study for abiotic stress tolerance.

Response: Thanks for your insightful comments, we added Figure 9 to illustrate the possible response pathways of BsMADS33 to ABA stress.

  1. All the figure quality is poor and very small. Therefore, it is tough to see the letter in the figure. Please enlarge all of them with high resolution.

Response: Thank you for your valuable advice. As Reviewer suggested that we have increased the resolution of all the figures. We also provide the original figures in the attachment, please check it.

  1. Enlarge figure 8 with portrait mode.

Response: Considering the Reviewer’s suggestions, we enlarge Figure 8 with portrait mode and high resolution.

  1. The introduction is very short and did not include many recent reported genome-wide studies. Please cite the following study: a. Genome-Wide Identification and Characterization of PIN-FORMED(PIN) Gene Family Reveals Role in Developmental and Various Stress Conditions in Triticum aestivum. b. Genome-wide identification and expression pattern analysis of the KCS gene family in barley. c. Genome-wide identification and characterization of abiotic stress-responsive lncRNAs in Capsicum annuum. d. Genome-Wide Identification and Characterization of the Brassinazole-resistant (BZR) Gene Family and Its Expression in the Various Developmental Stage and Stress Conditions in Wheat (Triticum aestivum). e. Genome-wide identification and functional characterization of natural antisense transcripts in Salvia miltiorrhiza. f. Genome-wide identification and expression analysis of the AT-hook Motif Nuclear Localized gene family in soybean.

Response: Thank you for your careful work. We are very sorry for our negligence of the latest genome wide articles, and have supplemented the studies in the introduction section of our paper.

Change at                                         

L72 deisgnated BsMADS01 to designated BsMADS01.

Manuscript is plagiarized at L32-35, L98-99, L110, L123-124, L128-129, L136-137, L147, L170-171, L290-291, L306-309, L333-334, L341-342, L351-352. Please clean it.

Response: We have re-written those passages according to the Reviewer’s suggestion.

Special thanks to you for your good comments.

We tried our best to improve the manuscript and made some changes in the manuscript. These changes will not influence the content and framework of the paper.

Once again, thank you very much for your comments and suggestions.
Yours sincerely,
Corresponding author: Zhe-zhi Wang and Jun-feng Niu
E-mail: zzwang@snnu.edu.cn; niujunfeng@snnu.edu.cn

Reviewer 2 Report

Ze-yuan Mi et al. reported the identification and characterization of MADS-box Genes in plant Bletilla striata. Totally 45 members were identified and classified into five subfamilies. Clues of dynamic gene regulation of members were provided by searching for cis-acting elements in the promoter region. Importantly, authors showed that expressions of certain family members were tightly regulated in a tissue and/or stress specific manner, indicating that these MADS transcription factors may play important roles in these biological processes. The experiments are carefully designed and performed at high standard. The manuscript is well written. I only have some minor concerns:

  1. The major goal and work of this study is the identification of MADS gene members in Bletilla striata. Authors should introduce their strategy and approach in very detail. The brief introduction in the method is not enough. Especially, the input sequence, parameters and cutoff that used in any program should be clearly shown and discussed for the reason of choice.
  2. The font size should be consistent across all figures (including supplemental figures). For example, gene labels and cis-acting elements names in figure2 are too small to be clearly read. Authors should carefully examine and correct all figures and make sure that they meet the requirement of Plants journal.
  3. In figure 8, authors performed RT-qPCR to study the gene expression of MADS family members under different stresses. Considering the different treatment condition, protocol, reagents, facility and time duration, the abiotic stresses can be very unstable and their effectiveness is a concern. As a result, it is safe for the author to show the efficiency and success of stress treatments. My suggestion here is that, under each stress condition, authors should select the marker genes (salt-responsive genes under NaCl condition, for example) in Bletilla striata and measure the marker gene’s expression (to make sure the treatment is effective), together with other MADS genes, under that condition.

Author Response

Dear  Reviewers:

On behalf of my co-authors, we thank you very much for giving us an opportunity to revise our manuscript, we appreciate editor and reviewers very much for their positive and constructive comments and suggestions on our manuscript entitled “Genome-Wide Analysis and the Expression Pattern of the MADS-box Gene Family in Bletilla striata” (ID: plants-1386932). Those comments are all valuable and very helpful for revising and improving our paper, as well as the important guiding significance to our research. We have studied comments carefully and made correction which we hope meet with approval. Revised portion are marked in yellow highlights in the paper. All changes could be found in the “manuscript-marked” file. The yellow highlights show the changes we made according to the reviewer’s comments. We would like to express our great appreciation to you and reviewers for comments on our paper and hope that the correction will meet with approval.

Reviewer 2

  1. The major goal and work of this study is the identification of MADS gene members in Bletilla striata. Authors should introduce their strategy and approach in very detail. The brief introduction in the method is not enough. Especially, the input sequence, parameters and cutoff that used in any program should be clearly shown and discussed for the reason of choice.

Response: Thank you very much for your professional advice, which is very important to us. We have polished the materials and methods section of our article.

  1. The font size should be consistent across all figures (including supplemental figures). For example, gene labels and cis-acting elements names in figure2 are too small to be clearly read. Authors should carefully examine and correct all figures and make sure that they meet the requirement of Plants journal.

Response: Thank you for your careful reading. As Reviewer suggested that we improved the quality of the figures.

  1. In figure 8, authors performed RT-qPCR to study the gene expression of MADS family members under different stresses. Considering the different treatment condition, protocol, reagents, facility and time duration, the abiotic stresses can be very unstable and their effectiveness is a concern. As a result, it is safe for the author to show the efficiency and success of stress treatments. My suggestion here is that, under each stress condition, authors should select the marker genes (salt-responsive genes under NaCl condition, for example) in Bletilla striata and measure the marker gene’s expression (to make sure the treatment is effective), together with other MADS genes, under that condition.

Response: Thank you for your instructive advices. The suggestions of Reviewers are very useful and provide us with good experimental ideas. We also considered that before our experiment, but Bletilla striata is a relatively small niche research species with weak molecular biology research foundation. This approach is based on the fact that there are genes have been experimentally verified to respond to stress, but we have not found any of this online. Through reviewing a large number of gene family articles, we learned that housekeeping genes were expressed stably and could be selected as reference genes for expression pattern analysis.  According to our laboratory studies, we selected BsGAPDH as the internal reference gene in this work.

Thank you for your hard work and professional comments

We tried our best to improve the manuscript and made some changes in the manuscript. These changes will not influence the content and framework of the paper.

Once again, thank you very much for your comments and suggestions.
Yours sincerely,
Corresponding author: Zhe-zhi Wang and Jun-feng Niu
E-mail: zzwang@snnu.edu.cn; niujunfeng@snnu.edu.cn

Round 2

Reviewer 1 Report

I am happy with the authors reply. Manuscript looks refine now and can be accepted.